# Neural networks can be FLOP-efficient integrators of 1D oscillatory integrands

**Anshuman Sinha**
*School of Computational Science & Engineering, Georgia Institute of Technology, Atlanta, GA*     *anshs@gatech.edu*

**Spencer H. Bryngelson**     *shb@gatech.edu*
*School of Computational Science & Engineering, Georgia Institute of Technology, Atlanta, GA*
*Woodruff School of Mechanical Engineering, Georgia Institute of Technology, Atlanta, GA*
*Guggenheim School of Aerospace Engineering, Georgia Institute of Technology, Atlanta, GA*

**Reviewed on OpenReview:** *https://openreview.net/forum?id=5psgQEHn6t*

## Abstract

We demonstrate that neural networks can be FLOP-efficient integrators of one-dimensional oscillatory integrands. We train a feed-forward neural network to compute integrals of highly oscillatory 1D functions. The training set is a parametric combination of functions with varying characters and oscillatory behavior degrees. Numerical examples show that these networks are FLOP-efficient for sufficiently oscillatory integrands with an average FLOP gain of $10^3$ FLOPs. The network calculates oscillatory integrals better than traditional quadrature methods under the same computational budget or number of floating point operations. We find that feed-forward networks of 5 hidden layers are satisfactory for a relative accuracy of $10^{-3}$. The computational burden of inference of the neural network is relatively small, even compared to inner-product pattern quadrature rules. We postulate that our result follows from learning latent patterns in the oscillatory integrands that are otherwise opaque to traditional numerical integrators.

## 1 Introduction

Numerical integration of highly-oscillatory functions is required for problems in fluid dynamics, nonlinear optics, Bose–Einstein condensates, celestial mechanics, computer tomography, plasma transport, and more (Connor & Curtis, 1982; Iserles, 2005). Classical numerical integration schemes are based on quadrature rules, like those of Newton–Cotes type (e.g., trapezoidal or Simpson's rule), Romberg integration, or Gauss quadrature (Davis & Rabinowitz, 2007; Milne, 2015; Hildebrand, 1987). These are unsuited for highly oscillatory integrands, requiring many quadrature points before reaching their asymptotic convergence rates. This work uses feed-forward, fully connected neural networks as approximate integrators for highly oscillatory integrands. Focus is paid to the floating point operation (FLOP)-efficiency of different integrators: Can a feed-forward neural network outperform classical integration schemes for a given FLOP budget?

## 2 Background

Numerical integral methods crafted for highly oscillatory integrands have been developed. These are, for example, based on the stationary phase approximations (Filon, 1930; Levin & Sidi, 1981; Levin, 1996; Iserles & Nørsett, 2005; Evans & Chung, 2007; Hascelik, 2009). Each method is powerful when used appropriately but operates under relatively strict conditions, including the type of oscillatory features (e.g., sine and cosine). These methods can be generalized when more sophisticated algorithms are added to the integration technique to identify a suitable basis for integration. Though the methods for this identification are not well documented

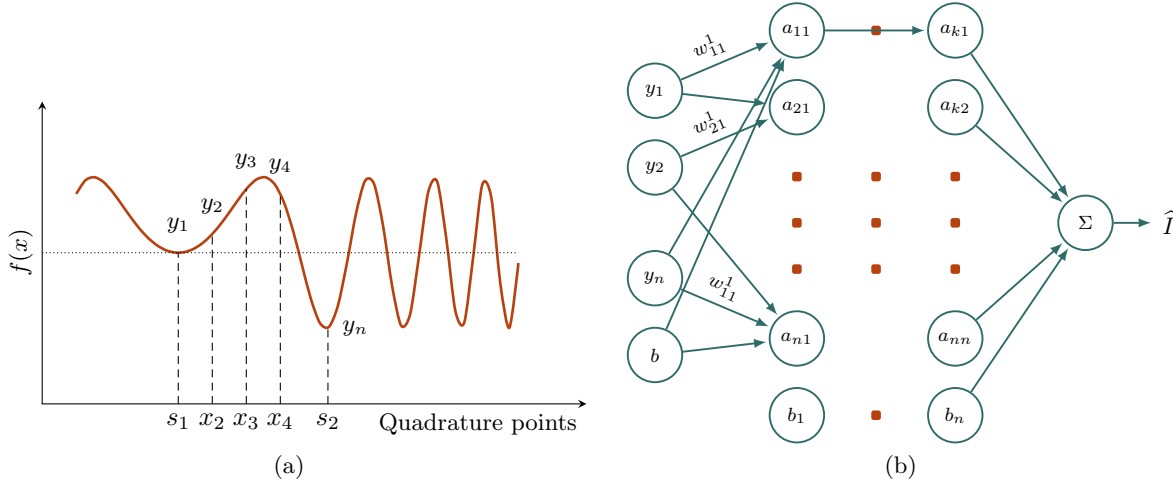

Figure 1: (a) Newton–Cotes-like method of approximating an integral with a (b) feed-forward multi-layer perceptron neural network (NN). The model uses inputs $f(x_i)$, where the pseudo-quadrature points $x_i$ are fixed in the spatial domain, to compute the weights and biases of the neural networks, thus approximating $\widehat{I}$.

in the literature or any open-source software, their use or evaluation of the additional computational cost associated with them is challenging to align with our objective.

More general techniques for dealing with highly oscillatory integrands are desirable. Neural networks are one such possibility for generalization. Indeed, neural networks have been used to approximate integrals before. Previous works used single hidden layer neural networks for PID controller applications (Zhe-Zhao et al., 2006), and dual neural networks with application to material modeling (Li et al., 2019). These neural networks were particularly useful in many-query settings but do not appear to generalize beyond their specific applications.

Some works have taken a tailored approach to neural network design for integration. Ying Xu & Jun Li (2007) used oscillatory basis functions for interpolation and integration, though the cosine activation function prevents approximation to broader function classes (Wu, 2009). Lloyd et al. (2020) trained single-layer networks for multidimensional integrals, focusing on parameterized, many-query problems, and showed promising results. The main limitation of Lloyd et al. (2020) is the restriction to high-dimensional many-query problems for FLOP-efficiency. This work investigates the FLOP-efficiency, or integration accuracy per required floating point operation, for integrating *1D functions* via a feed-forward fully connected neural network. We aim to provide a computationally efficient solution for problems that require repeated integration of an integrand with varying parameters.

## 2.1 Problem definition

The formulation is to compute the integral $I$ of any function $f(\boldsymbol{x})$ in a bounded domain $[s_1, s_2]$ which is expressed as follows,

$$I = \int_{s_1}^{s_2} f(\boldsymbol{x})\mathrm{d}\boldsymbol{x}. \tag{1}$$

The method of fig. 1, where the inputs of the network are shown as $f(x_i)$ for a fixed set of $x_i \in [s_1, s_2]$, and the outputs of the network are the integral values $I$ of the input function.

We train the neural network model with parametrically varying samples. These neural network integrators are compared with the classical numerical integration methods like Newton–Cotes quadrature rule, as shown in eq. (2). We do not consider more complex integration schemes like Gauss quadrature as they do not reach asymptotic convergence behaviors for realistic numbers of quadrature points and typically become unstable before this.

The input domain $[s_1, s_2]$ is divided into $n_q$ quadrature points. The integral is approximated as a weighted sum of function values at those quadrature points $(x_i)$.

$$\int_{s_1}^{s_2} f(\boldsymbol{x})\mathrm{d}\boldsymbol{x} \approx \sum_{i=1}^{n_q} w_i f(x_i) \tag{2}$$

Common Newton–Cotes formulations include the second-order accurate trapezoidal method

$$\int_{s_1}^{s_2} f(\mathbf{x})\mathrm{d}\boldsymbol{x} \approx \sum_{k=1}^{n_q} \frac{f(x_k) + f(x_{k+1})}{2}\Delta x, \tag{3}$$

where $x_k$ are uniformly spaced points in the domain $x \in [s_1, s_2]$ and $\Delta x = (s_2 - s_1)/n_q$, the first-order accurate is the mid-point method

$$\int_{s_1}^{s_2} f(\mathbf{x})\mathrm{d}\boldsymbol{x} \approx \sum_{k=1}^{n_q} f\left(s_1 + \left(k - \frac{1}{2}\right)\Delta x\right)\Delta x. \tag{4}$$

and the third-order accurate Simpson's method

$$\int_{s_1}^{s_2} f(\mathbf{x})\mathrm{d}\boldsymbol{x} \approx \sum_{k=1}^{n_q/2} \frac{(f(x_{2k-2}) + 4f(x_{2k-1}) + f(x_{2k}))}{3}\Delta x, \tag{5}$$

where, again, $f(\cdot)$ is the integrand of interest.

## 2.2 Neural network method

The network's learned weights reduce computational costs. With an optimized architecture, we seek accurate integration using few input (quadrature) points. Improved accuracy for a given number of quadrature points is feasible as the neural network has trainable weights for approximation, but classical techniques rely on fixed interpolating functions. For such methods, the more oscillatory a function, the more quadrature points are required to achieve integration of the same accuracy.

In our notation, the values $y_i$ of an integrand $f(\boldsymbol{x})$ at abscissa $x_i$ , $(y_i = f(x_i)\ \ i \in 1, \ldots, n)$ are shown in fig. 1. The parameters

$$a_{ij} = \sigma\left(\sum_{i=1}^{n} w_{ij}x_{ij} + b_i\right). \tag{6}$$

form each layer of fig. 1, $x_{ij}$ represent the information from previous layers (i.e, $a_{(i-1)j}$) and $b_i$ is the bias term in each layer. In the final layer of fig. 1, the sum of each input to a final node ($\Sigma$) and the output $\widehat{I}$ to train the network via back-propagation. ReLU is used as the activation function. The number of hidden layers and the number of neurons are optimized during hyperparameter optimization. Section 3.4 shows the results for the hyperparameter optimization for the model.

# 3 Experiments

We evaluate the proposed neural network method with several example cases. We present the employed dataset, metric of evaluation, and other details on the hyperparameter study, then the results.

## 3.1 Experimental dataset

### 3.1.1 Training data

The model inputs the function values and the baseline integral value as output in the current work. The training is done by varying the function's parameters while keeping its nature the same. While training for

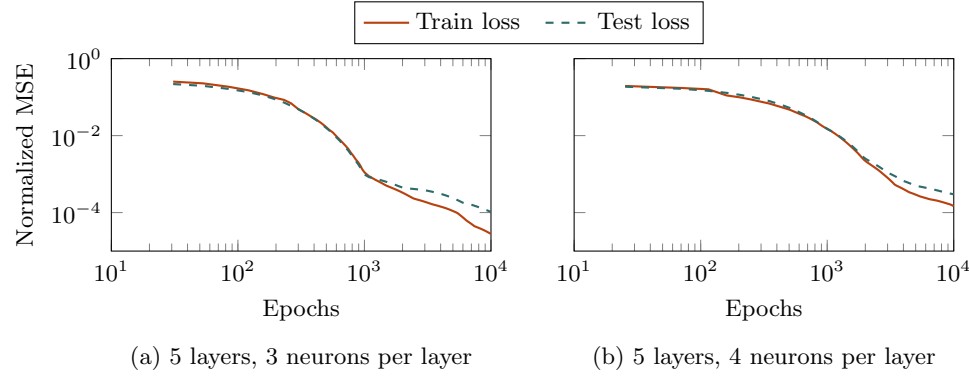

(a) 5 layers, 3 neurons per layer    (b) 5 layers, 4 neurons per layer

Figure 2: Neural network (NN) model's performance on test and training dataset.

one network configuration of the model, we keep the number of quadrature points fixed. We trained separate models for numbers of inputs ranging from $2^0$ to $2^{13}$ and evaluated their performance on separate test data. These inputs (with varying quadrature points) were tested on the specific network configuration, yielding results with varying validation accuracy. For a fixed number of quadrature points, the input values of the model are the function's values $f(x_i)$ at those quadrature points $x_i$.

### 3.1.2 Testing data

The model is tested by calculating the integral of a function that is the same kind as the training functions but parametrically different and unseen. The integral is calculated within the same input space, i.e., within the same limits of the input domain $[s_1, s_2]$. The training data are divided into disjoint subsets, with an 80–10–10 split for train, test, and validation.

### 3.2 Evaluation

We evaluate the performance of the neural network model using the normalized mean squared error (Normalized MSE) metrics on the test set. The error in the model's result $(\widehat{I}_k)$ is computed against a representative exact solution to the integral $(I_k)$, which is evaluated using the trapezoidal method with $2^{13}$ quadrature points. This strategy is sufficiently accurate for our purposes, which is checked against more and fewer such baseline quadrature points. We use the surrogate truth integrand $I_k$ for the $k$'th sample in the test set as

$$I_k = \sum_{j=1}^{n_q=2^{13}} \left( \frac{f(x_j) + f(x_{j+1})}{2} \right) \Delta x, \tag{7}$$

where $n_q = 2^{13}$ for $\Delta x = (s_2 - s_1)/2^{13}$ while calculating the integral of $f(x)$ (the integrand) between $[s_1, s_2]$ using the trapezoidal rule in eq. (3).

The normalized mean–square error (MSE) for the test set is calculated using the $m$ samples of functions is

$$\text{Normalized MSE} = \frac{1}{m} \sum_{k=1}^{m} \frac{(I_k - \widehat{I}_k)^2}{I_k^2}. \tag{8}$$

These $m$ samples are generated by parametric variations of the same function, as described in table 1. We evaluate the performance of the neural network model by comparing it to standard numerical integration methods, such as the trapezoidal and midpoint methods.

### 3.3 Hyperparameter optimization

The neural network's architecture was optimized according to a test-set error value below $10-3$ and minimizing a fixed number of FLOPs. The validation data are used for hyperparameter optimization. Table 1 shows the optimization results.

Table 1: Result of the neural network model on various oscillatory functions. The normalized FLOP gain for a given accuracy is $\alpha$, which serves as a measure of performance (higher is better) and is computed with respect to the classical quadrature-based model as shown in eq. (9). All functions are 1D in $\boldsymbol{x}$.

| Function Type | Equation | Parameter space | $\alpha$ | Result |
|---|---|---|---|---|
| Bessel$(k, \nu)$ | $\cos(k\boldsymbol{x})J_0(\nu, \boldsymbol{x})$ | $\nu \in [125, 175],\ k \in [75, 125]$ | 6.01 | Figure 4 (a) |
| Evan–Webster-1$(k_1, k_2)$ | $\cos(k_1\boldsymbol{x}^2)\sin(k_2\boldsymbol{x})$ | $k_1 \in [5, 15],\ k_2 \in [25, 75]$ | 17.72 | Figure 4 (b) |
| Rayleigh–Plesset$(\rho)$ | See appendix A.3 | $\rho \in [500, 1000)$ | 23.46 | Figure 4 (c) |
| Evan–Webster-2$(k)$ | $\exp(\boldsymbol{x})\sin(k\cosh(\boldsymbol{x}))$ | $k \in [25, 75]$ | 19.60 | Figure 4 (d) |
| Sine$(k)$ | $\sin(k\boldsymbol{x})$ | $k \in [5, 15]$ | 0.91 | Figure 6 (a) |
| Exponential$(k)$ | $\exp(k\boldsymbol{x})$ | $k \in [1, 5]$ | 0.60 | Figure 6 (b) |

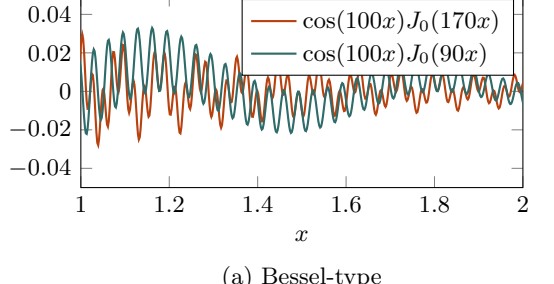
(a) Bessel-type

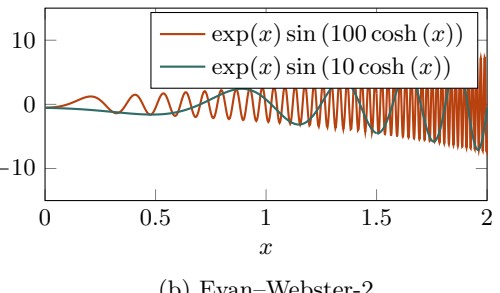
(b) Evan–Webster-2

Figure 3: Example canonical oscillatory test functions as labeled.

The architecture was optimized heuristically with an iterative increment of network hidden layers and neurons in each layer. Each configuration was executed using the model until the training normalized MSE stagnated. An example of this convergence is shown in fig. 2, along with the testing and training performance for two separate network configurations. The performance of each network configuration was evaluated on a validation dataset, which was separate from the training data. The iterative tests were done with increasing samples as the parameters of the network increased (Bishop, 1995). The upper limit complexity of the network is obtained via the performance of the classical integrators. With the increasing complexity of the network, we observe diminishing returns against the FLOP burden. A deeper neural network can still achieve smaller integration errors. We aim to achieve the highest accuracy for a given computational expense while avoiding overfitting.

### 3.4 Results

The results of this study present the current model's applicability for efficiently computing and predicting the integrals of oscillating functions. The model is tested on oscillatory and non-oscillatory functions.

Table 1 compares results for various functions with different oscillatory features. The degree of *oscillatoriness* is defined by the function's parameters, which are denoted in parentheses in the "Function Type" column of table 1. A larger coefficient corresponds to a more oscillatory function. The parameter space column is the range in which the function's parameters vary to create the training data. The gain in the number of FLOPs, shown in eq. (9), defines the computational advantage obtained for the number of required floating point operations while implementing the current method over the other numerical-based methods of computing the integral. The term $\text{FLOPs}_{\text{NN}}$ represents the number of FLOPs required by the neural network model, and $\text{FLOPs}_{\text{QM}}$ represents the number of FLOPs required by a classical integration quadrature based model.

The number of FLOPs for the neural network method is calculated by enumerating the floating-point operations involved in computing the output of each neuron in every layer. This computation follows the

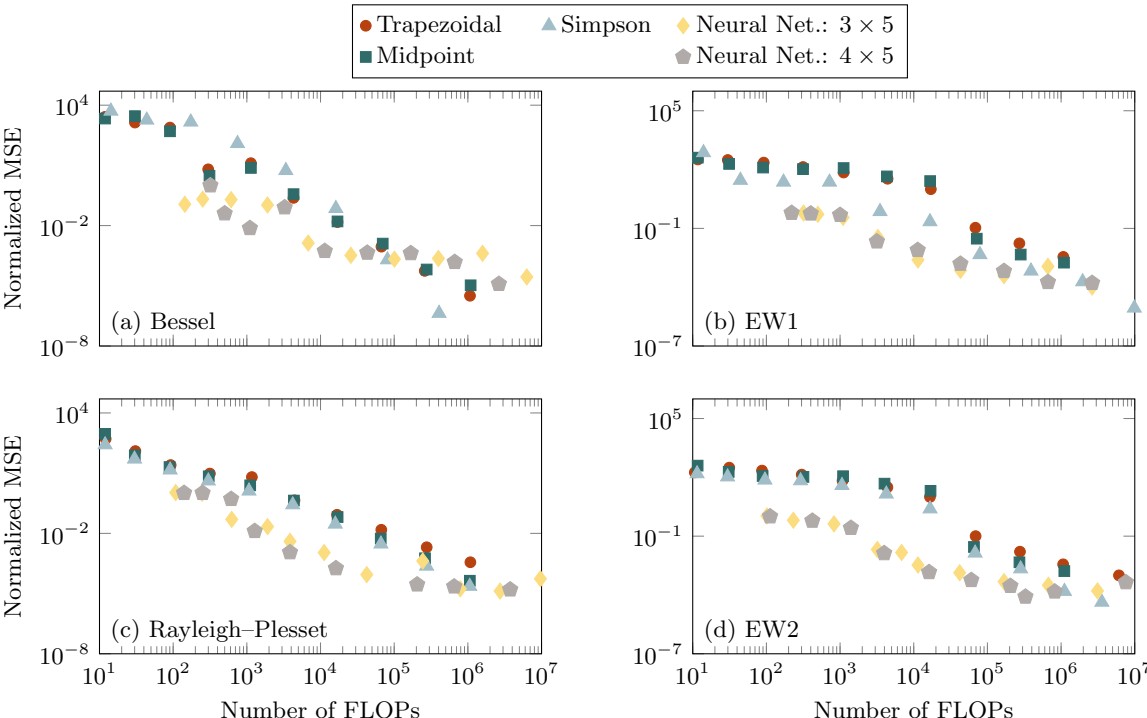

Figure 4: Comparison of results of neural network (NN) model with Newton–Cotes methods for the oscillatory functions of table 1. The integrands associated with panels (a) and (b) are as shown in fig. 3.

formula given in eq. (6). For an $H$ layer fully-connected feed-forward neural network with each layer having $N$ neurons, each activation $a_{ij}$ comprises $N$ multiplications and addition operations. Given $N$ operations per layer and $H$ layers, the total FLOP count is $(4N + 2)N^2H(1 + n_q)$. The number of FLOPs associated with the trapezoidal, mid-point, and Simpson's methods are $2n_q + 1$, $3n_q + 1$ and $4.5n_q + 2$, following eqs. (2), (4) and (5).

To study the influence of the oscillatory nature of the input function on the results, we performed integral calculations for the function while gradually increasing its oscillatory behavior. We tested the model on sinusoids with varying frequencies (increasingly oscillatory). The measure of neural network integration efficiency is the normalized ratio of FLOP gain for a neural network model over traditional integration for the same MSE error, $\alpha$:

$$\alpha = \frac{|\text{FLOPs}_{\text{NN}} - \text{FLOPs}_{\text{QM}}|}{\text{FLOPs}_{\text{NN}}}. \tag{9}$$

A larger $\alpha$ corresponds to a higher normalized gain by the neural network model.

fig. 3 shows the results of the model's performance by providing examples of individual functions. fig. 4 shows the test set's normalized mean square error (NMSE) in integral computation from the model as a function of the number of FLOPs required to compute the individual integral. Figure 4 shows the results of normalized MSE loss values for different numbers of quadrature points as input training data points for the model.

Table 2 shows the results for the two best-performing network configurations (neurons × hidden layers) achieved through parametric optimization. The number of FLOPs required for computing the integral increases as the number of quadrature points increases. Increasing quadrature points also increases the accuracy of the integral computation for both methods. For a normalized MSE of $10^{-3}$, the neural network strategy computes the integral using fewer FLOPs than traditional quadrature methods, making it FLOP-efficient.

We observe the opposite result for the less oscillatory functions of fig. 5. Figure 6 shows that the neural network model requires more FLOPs to compute the same integral for a given normalized MSE. This result

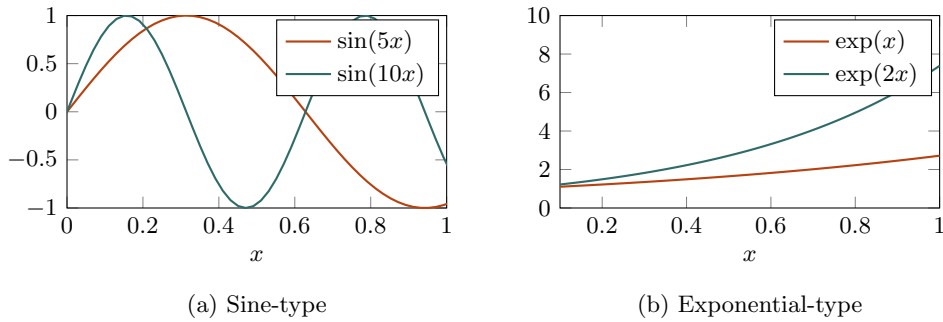

Figure 5: Example test functions tested as labeled.

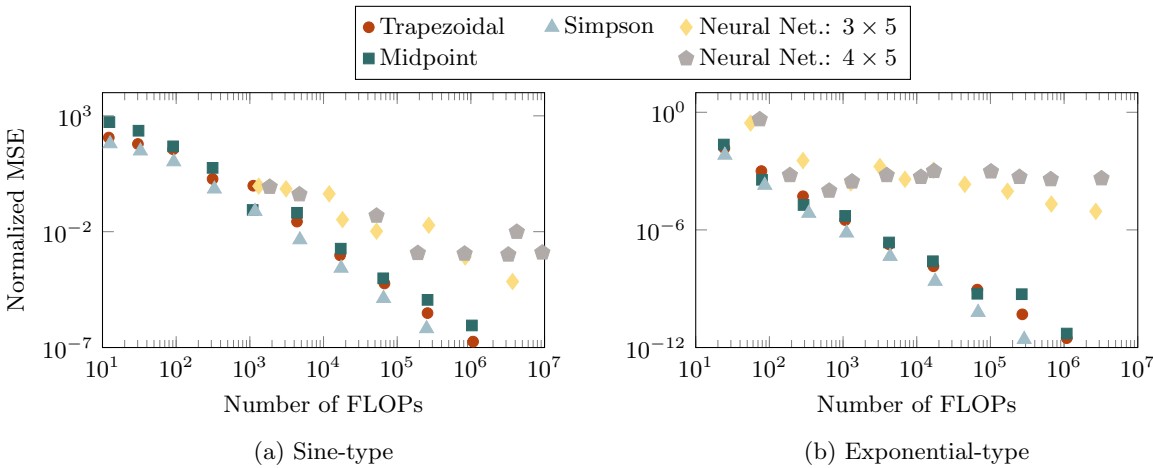

Figure 6: Results of the NN model with traditional numerical integration methods for non-oscillatory (a) sine and (b) exponential functions shown in fig. 5.

Table 2: Model hyperparameter selection space. The optimum values are chosen based on the normalized MSE. The sample number, learning rate, and test–train split are fixed.

| Hyperparameter | Search space | Optimum value |
|---|---|---|
| Number of hidden layer | $\{1, 2, 3, 4, 5\}$ | 3 Hidden layers |
| Neurons in each hidden layer | $\{1, 2, 3, 4, 5, 6, 7\}$ | 5 Neurons |
| Number of samples | $\{10^2, 10^3, 10^4\}$ | $10^4$ Samples |
| Learning rate | $\{10^{-5}, 10^{-4}, 10^{-3}\}$ | $10^{-4}$ |
| Test-train split | $\{0.1, 0.15, 0.2\}$ | 0.15 |

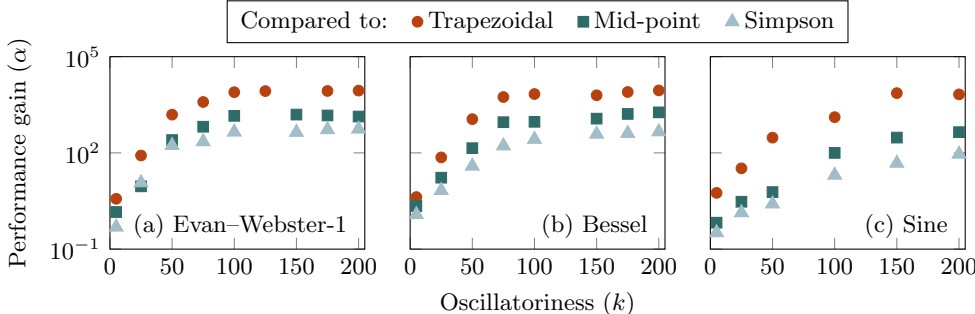

Figure 7: Performance gain via the neural integrator for increasingly oscillatory integrands as labeled which is parametrically increased according to their parameters shown in table 1.

is expected, as these integrands can be computed accurately with only few quadrature points using low-cost Newton–Cotes methods.

Figure 7 shows that the performance gain increases rapidly with the increasing oscillation of the integrand. This indicates that the numerical integral methods require more quadrature points to evaluate the same integral. After some level of oscillation has been reached, the performance gain curve starts to plateau. Even if one adds more oscillations to the domain (by increasing the oscillatory parameters), the integral value within the sub-domain is the same. Thus, no further gain in performance is expected. For less oscillatory integrands, the values of $\alpha$ decay to unity. The neural network integral then requires more FLOPs than a classical integrator.

The summary of the comparative study on various functions is presented in table 1. The number of FLOPs for this comparison was calculated for a fixed Normalized MSE value of $10^{-3}$ for the loss function. This value was chosen to remain within the application limits for downstream integration use. Overall, section 3.4 shows a general trend of decreasing normalized MSE in the integral as the complexity of the network increases. The neural network model predicts the integral result with the same accuracy but exhibits nearly two orders of magnitude better efficiency.

## 4    Discussion and conclusion

An approach for computing the integrals of 1D functions of highly oscillatory and non-oscillatory behaviors. We used a feed-forward neural network to estimate the integrals and evaluated their accuracy for a fixed FLOP budget. In a comparative study, several cases of oscillatory functions were examined. The approach of calculating integrals with a neural network model outperforms existing numerical methods for a fixed FLOP budget. The neural network model was an increasingly efficient integrator for functions with more oscillatory behavior. The proposed method does not extrapolate to qualitatively different integrands than it was trained on, mostly owing to the restriction of FLOP efficiency. Thus, our results are most applicable to cases where parametric variations of an integrand must be evaluated in a many-query setting, and one can forecast the integrand characteristics.

### Acknowledgement

The authors appreciate discussion with Dr. Ethan Pickering at an early stage of this work. SHB acknowledges support of the US Office of Naval Research under grant no. N00014-22-12519 (PM Dr. Julie Young).

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

## A Appendix

### A.1 Neural network memory costs

For a fully connected neural network with $L$ layers of $N$ neurons, each with a 4-byte floats parameterization, the memory footprint in bytes, ignoring input layer biases, is

$$4[N^2(L-1) + N(L-2)], \tag{10}$$

for $N^2(L-1)$ weights, $N(L-2)$ biases (each neuron in the hidden and output layers, excluding the input layer), and 4 bytes per float. Thus, the memory footprint for a 5-layer, 3-neuron network is 180 bytes, which is negligible in almost any practical computing environment.

### A.2 Training dataset generation

The general data generation process for training is

$$y(\boldsymbol{x}) = f(k, \boldsymbol{x}), \tag{11}$$

where $y(\boldsymbol{x})$ is the function generated for a set of $\boldsymbol{x}$ values with parameter $k$. The total number of $\boldsymbol{x}$ for a simulation defines the total number of samples. Here, $k$ is the randomly selected parameter associated with the oscillatoriness of the integrated for each $\boldsymbol{x}$, $k \in [k_1, k_2)$.

| Param. | Value | Description | Units |
|--------|-------|-------------|-------|
| $\Delta p$ | $-7670$ | Ambient pressure difference | Pa |
| $p(t)$ | $1.3 \times 10^6 \cos(53000\pi t)$ | Driving pressure | Pa |
| $\sigma$ | $0.0725$ | Surface tension | N/m |
| $\mu$ | $8.9 \times 10^{-4}$ | Dynamic viscosity | Pa s |
| $k$ | $1.33$ | Polytropic exponent | – |

Table 3: Parameterization of the Rayleigh–Plesset equation.

### A.3 Rayleigh–Plesset dynamics

The Rayleigh–Plesset equation represents the oscillatory radial dynamics of gas bubbles suspended in a liquid. The equation is a second-order ODE and its solution is highly oscillatory and dominated by nonlinear behavior in many regimes. Specifically, it can be expressed as

$$\rho \left( R\frac{\mathrm{d}^2 R}{\mathrm{d}t^2} + \frac{3}{2}\left(\frac{\mathrm{d}R}{\mathrm{d}t}\right)^2 \right) = \Delta p - p(t) - \frac{2\sigma}{R} - \frac{4\mu}{R}\frac{\mathrm{d}R}{\mathrm{d}t} + \left(\frac{2\sigma}{R_0} - \Delta p\right)\left(\frac{R_0}{R}\right)^{3k} \tag{12}$$

$$R(0) = R_0, \ \frac{\mathrm{d}R}{\mathrm{d}t}(0) = 0,$$

where the bubble radius $R$ is the dependent variable and time $t \in [0, T]$ is the independent variable. Equation (12) is buttressed by initial conditions for bubble radius $R_0 = 2.6 \times 10^{-6}$ (all following quantities in SI units) and zero initial radial velocity of the spherical bubble. The coefficients in eq. (12) are expressed in table 3. Elaborating on their meaning: the ambient pressure difference represents the initial discontinuity between the pressure inside the bubble and the pressure in the suspending liquid, the driving pressure is how the pressure of the liquid surrounding the bubble changes in time, the surface tension coefficient is a standard quantity representing the proclivity for one fluid to want to adhere to another, the dynamic viscosity is the resistance of the suspending liquid to changes in motion due to a surrounding change in velocity, and the

polytropic coefficient serves as a mediating rate for which the gas inside the bubble is able to expand or contract.

We generate an arbitrarily varying set of oscillatory data by varying the ambient density of the suspending liquid as $\rho \in [500, 1000)$. Equation (12) is integrated over the time interval, which is accomplished usiing a traditional numerical differential equation solver with a suitably small time step that captures the short time scales occurring at solutions cusps (corresponding to bubble size decay to growth). The solution is oscillatory in the regimes tested, but integrals of it are required in the context of underwater hydrodynamics broadly (Bryngelson et al., 2023; Bryngelson & Colonius, 2020; Plesset & Prosperetti, 1977).

