# OpenReview forum: "Neural networks can be FLOP-efficient integrators of 1D oscillatory integrands"
_TMLR — Accepted by TMLR_

### Review · Reviewer_vUXA · 2024-02-06

**Summary Of Contributions:**

This paper aims to train neural networks to predict the integral of oscillatory functions. The input of the model are grid-point values of the underlying function, the output is the value of the integral of this function.

The authors consider different families of such oscillatory functions, each parametrized by one or two parameters. An example is $sin(kx)$ with parameter $k$. They argue that (fully connected) neural networks can be flop efficient compared to standard numerical quadrature methods (Newton-Cotes).

**Audience:**

No

**Claims And Evidence:**

No

**Requested Changes:**

* For this submission to be of interest to a wider readership, the experiments would have to be considerably extended, beyond the simple function families considered in this work. For the Sine-family, Figure 5(a) suggests that the family is actually not very oscillatory. How far can one push the values of $k$?
* An important question that is not addressed is whether such models, trained on a simple model class, can generalize beyond this model class. This would be crucial for practical applications.
* I believe that with the specific integer-valued parameters considered by the authors, the oscillatory integrands lead to very regular integrals as a function of the parameters. The authors should include a plot of the integral $I=I(k)$ or $I=I(k_1,k_2)$ as a function of $k$ and $(k_1,k_2)$ for the various families, respectively.
* I suggest, the authors should extend their baselines; Would the trained NN be FLOP-efficient compared to Simpson's rule?

**Strengths And Weaknesses:**

Strengths
* The general question, whether standard numerical methods can be improved/complemented by novel tools from ML is interesting. Quadrature is a core task in many numerical problems, hence there might be interest in such "learned quadrature rules".

Weaknesses
* Variety of datasets: The authors consider only 4 very special sets of functions, and from my reading of Table 1, these classes each only contain the following number of samples:
    - 101 (class "Bessel")
    - 11*51 = 521 (class "Evan-Webster")
    - 11 (class "Sine")
    - 5 (class "Exponential")

    Based on experiments with such limited datasets, it is not clear, whether/in which applications such a model could be practically useful. It is not clear whether the learned model generalizes beyond its (narrow, i.e. 1- or 2-parameter) trained class. As such, I would question whether this work is of general interest to the TMLR readership.
* Baseline: The authors compare accuracy vs. FLOPs, to argue that neural networks can outperform standard numerical quadrature methods. However, the baseline quadrature here are the trapezoidal and mid-point rules, which are relatively low-order.

---

> ### Author Response · Authors · 2024-03-02
> **Response to reviewer vUXA**
>
> ## Weaknesses
>
> 1. Variety of datasets: The authors consider only 4 very special sets of functions, and from my reading of Table 1, these classes each only contain the following number of samples [...]
>
>    __Response:__  We have added more classes of functions, but also note that there was a mistake in our notation in Table 1. We actually randomly sample over the range of the "oscillatory" parameter(s) for training and testing. We apologize for the mistake and thank the referee for catching it.
>
> 2. Baseline: The authors compare accuracy vs. FLOPs, to argue that neural networks can outperform standard numerical quadrature methods. However, the baseline quadrature here are the trapezoidal and mid-point rules, which are relatively low-order.
>
>    __Response:__  We have added Simpson's higher-order accurate quadrature rule to all our experiments. Results show no meaningful difference between it and the other methods. This can be anticipated: No quadrature method of Newton-Cotes-type, Gauss-type, or otherwise will be in its asymptotic convergence regime for sufficiently highly oscillatory integrands. In fact, this idea motivated some of the work from the start!
>
> ## Requested Changes
>
> 1. For this submission to be of interest to a wider readership, the experiments would have to be considerably extended, beyond the simple function families considered in this work. For the Sine-family, Figure 5(a) suggests that the family is actually not very oscillatory. How far can one push the values of?
>
>     __Response:__ We have added new figures to show examples of other highly oscillatory integrands, including some that have thousands of oscillations over the integral domain. We have also further explained our sine-family experiments in the text to clarify some of this. Thanks!
>
> 2. An important question that is not addressed is whether such models, trained on a simple model class, can generalize beyond this model class. This would be crucial for practical applications.
>
>     __Response:__  This is a very important question we should have addressed from the start. The referee's likely anticipated response is correct here: this strategy does _not_ generalize well outside of its training regime. This is because FLOP-efficiency is hard to achieve, and neural networks are not particularly good extrapolators. While this would be a nice feature, we argue that this is not of first-order importance for this manuscript. The reason is that many problems of scientific interest repeatedly approximate integrals of the same type millions, billions, or more times. Some straightforward examples are PDE solves, which can entail integral solves as a sub-step of the right-hand side evaluation procedure. An alternative to using a neural network in such situations is to tabulate accurate values of the integral. This is common practice in some cases but does not apply to oscillatory integrands because the value of the integral is highly sensitive to the integrand itself. A discussion of this has been added to the manuscript.
>
> 3. I believe that with the specific integer-valued parameters considered by the authors, the oscillatory integrands lead to very regular integrals as a function of the parameters. The authors should include a plot of the integral or as a function of and for the various families, respectively.
>
>    __Response:__  Similar to our response to the earlier point by reviewer vUXA, we added new figures to show examples of other highly oscillatory integrands, into the regime of hundreds or thousands of oscillations over the domain. Good suggestions.
>
> 4. I suggest, the authors should extend their baselines; Would the trained NN be FLOP-efficient compared to Simpson's rule?
>
>     __Response:__ Following our earlier response, we added Simpson's rule, which is higher-order accurate, for all of our experiments. We see no meaningful difference in results. Thus, none of the standard methods are in their asymptotic convergence regimes, so moving to a higher-order accurate method does not improve results for quadrature rules.

---

### Review · Reviewer_Eijb · 2024-02-16

**Summary Of Contributions:**

The authors propose to compute integral using a fully connected neural network. More specifically, the method consists is as follows:
1. For a selected family of functions, one gathers a dataset of the form $\\left\\{\\left(f_i(x_1),\\dots,f_i(x_N)\\right), I[f]\\right\\}$ where $I[f] = \int_{a}^{b}dx\\,f(x)$.
2. For this dataset, a feedforward neural network is trained to predict $I[f]$ from discrete values $\left(f_i(x_1),\\dots,f_i(x_N)\right)$
3. The resulting network is applied to unseen functions from the family.

The approach is tested on integrals for highly oscillatory functions, and it is shown that a trained neural network can evaluate integrals more efficiently (less FLOP for a given accuracy) than classical low-order methods such as trapezoidal and midpoint rules.

**Audience:**

Yes

**Claims And Evidence:**

Yes

**Requested Changes:**

1. Motivation is not clear.

Since, for the proposed approach, one needs to collect a dataset for each new family of functions (also, see 4. below), the scope of applications is not entirely clear. We kindly ask the authors to expand the discussion of the applications and/or the formulation of the problem (e.g., loss minimization over a distribution of functions). Note that for the classical methods, the situation is clear: once they are derived, it is possible to apply them to many different problems. The situation is quite different for neural networks, so it warrants a more detailed explanation.

---
2. It is questionable that the benchmarks are realistic.

It is not entirely clear, but it seems the method is tested on four families of functions (page 5, Table 1): "Bessel type," "Evan–Webster type," "Sine," and "Exponential." The spaces of parameters for these functions are small, e.g., for "Bessel type," authors listed $100$ distinct parameters, and for "Sine," only $10$. It is not clear whether the families used are broad enough. It is also not clear which functions the authors actually used since Figure 3 and Table 1 (both on page 5) contain conflicting descriptions and the code is not available.

I suggest authors to include:
1. discussion on the complexity of datasets, the frequency content of the functions included
2. tests on higher frequencies where asymptotic methods shine
3. more accurate description of the functions and their parameters used to generate datasets

---
3. Baselines are too weak.

The authors mention in the article that classical numerical schemes are not suitable. (page 1, Introduction)

> Classical numerical integration schemes are based on quadrature rules, like those of Newton–Cotes type: the trapezoidal and Simpson’s rules, Romberg integration, and Gauss quadrature (Davis & Rabinowitz, 2007; Milne, 2015; Hildebrand, 1987). These are unsuited for highly oscillatory integrands, requiring many quadrature points before reaching their asymptotic convergence rates.

Still, after that, the only baselines used are the Trapezoidal and Mid-point methods. If the methods are unsuitable, why use them as baselines? I agree that special integration techniques developed for highly oscillatory functions are not generally applicable, i.e., suitable only for particular parametric integrals. Still, I see no reason to discard them as baselines. For example, [1] contains methods applicable to a broad family of highly oscillatory functions.

---
4. No generalization study is available.

Arguably, the method is only useful, if it works for a broad class of problems after training on a small subset of problems. The generalization study is absent from the article. The study I have in mind may include answers to the following questions:

1. Suppose for particular functions we trained NN to approximate integrals of functions with oscillation frequencies $[w_1, w_2]$. What if we test the network on lower or higher frequencies? How does relative error depend on the difference between the frequency content of train and test sets?
2. Suppose we train on one family of functions and evaluate on another family of functions. What is the relative error of integration?
3. How does the relative error depend on the number of samples in the training set?

etc

---
5. Inductive bias of the model does not seem to be correct.

Integral is a linear operation, i.e., $\\int\\,dx\\left( f_1(x) + f_2(x)\\right) = \\int\\,dx f_1(x) + \\int\\,dx f_2(x)$. When a neural network is used to approximate the integral (ultimately, this is related to interpolation) there are at least two options:

1. $\\text{NN}(f) \simeq I[f]$
2. $\\text{NN}(f) = w,\\,I[f] = \sum_{I}w_i f(x_i)$

Option 2. has correct inductive bias, while option 1. leads to the approximation of a linear operator with a nonlinear one. I suggest authors discuss why they use 1. and not 2. or, better, to compare the accuracy and FLOPs of 1. and 2. Note that one can enforce consistency with 2. as explained in [2] (for interpolation problem).

---
6. Inaccuracies.

Below is a list of some minor problematic parts.

1. (page 3) Part below is not supported by experiments or theory.

> The network’s layers will approximate an intrinsic structure in the highly oscillatory integrands. This way, it decomposes the function’s highly oscillatory and complex nature into sub-functions that can be easily interpolated. This way, we can find the best-fitting integral with fewer data points, i.e., fewer input points.

2. (page 4) Normalized mean squared error is unusual. As a rule one consider average $\\frac{1}{m}\\sum\_{k=1}^{m} (I\_k - \\hat{I}\_{k})^2\\big/ I_k^{2}$.
What is the reason authors used  $\\sum\_{k=1}^{m} (I\_k - \\hat{I}\_{k})^2\\big/ \\sum\_{k=1}^{m}I\_k^{2}$?

3. (page 5) Figure 3 is not clear:

    1. It does not agree with Table 1 in the definition of "Bessel-type" and "Evan–Webster-type" since expressions in the legends are not the same as expressions given in the table.

    2. What is the $f(cos(x))$ in the legend for "Bessel-type"?

4. (page 6) The number of FLOP for NN seems to be incorrect. If I understood the architecture, the main computations occur in the first layer since the number of points used can be up to $2^13$. This is not reflected in the expression which gives the number of operations in terms of neutrons in hidden units.

---
7. Not reproducible.

Code and datasets or functions used to generate them are not available.

---

[1] -- S. Olver, Numerical Approximation of Highly Oscillatory Integrals.

[2] -- Y. Bar-Sinai, et al, Learning data-driven discretizations for partial differential equations.

**Strengths And Weaknesses:**

# Strengths

1. The idea is well articulated, and the proposed approach is clearly explained.
2. Explanation of experiments is informative.
3. Most claims are well supported.

# Weaknesses

1. Motivation is not clear.
2. It is questionable that the benchmarks are realistic.
3. Baselines are too weak.
4. No generalization study is available.
5. Inductive bias of the model does not seem to be correct.
6. Inaccuracies.
7. Not reproducible.

---

> ### Author Response · Authors · 2024-03-02
> **Response to reviewer Eijb**
>
> 1. Motivation is not clear
>
>     __Response:__  We agree that the motivation was not emphasized to the degree it should have been. We have appended the manuscript to address this. In particular, we point out that many scientific applications require repeated computation of integrals of the same type but varying parametric differences. These are so-called many query problems. By training a neural network that can approximate these integrals more efficiently than a classical integrator for, by our demonstration, all highly-oscillatory integrals tested (for which we added several to the manuscript), we provide the community with a tool to accelerate their scientific computations.
>
> 2. It is questionable that the benchmarks are realistic
>
>     __Response:__ We agree that more benchmarks are required. We have tackled two things to address this weakness and added them to the revised manuscript. First, we added other oscillatory functions and tested against them for a broad range of parameterizations. All showed results that were meaningfully better than classical integrators. This is shown in the new figures. We also clarified a mistake in the previous Table 1: the neural network is not trained on only a rather small array of parameters of the integrand. We randomly sample a broad range of parameters such that the training set is complete. Further, we have added results for different integrands that show the performance for varying oscillatoriness. We hope this comprehensively addresses the reviewer's concerns on this point!
>
> 3. Baselines are too weak
>
>     __Response:__  We have Added Simpson's method as a comparison for all the plots. It performs nearly the same as the midpoint and trapezoidal rule. This is because no classical integrator reaches its asymptotic regime (in terms of decreasing approximation error) with so few quadrature points over such a highly oscillatory integrand. Indeed, this motivated much of our work! We also note that while some approaches have been proposed for a class of integrands, they are not fully general nor provide code for comparison. They also do not measure FLOP efficiency, which is of critical interest when computing an integral millions of times (or perhaps many more!) in such a many-query setting.
>
> 4. No generalization study is available:
>
>     __Response:__ This is an important question raised by the referee. Indeed this strategy does not generalize well outside of its training regime. This is because FLOP-efficiency is hard to achieve, and neural networks are not particularly good extrapolators. However, this is not of primary importance for the objective of this manuscript: Many scientific computing problems repeatedly approximate integrals of the same type millions of times or more. Some straightforward examples are PDE solves, which can entail integral solves as a sub-step of the right-hand side evaluation procedure. An alternative to using a neural network in such situations is to tabulate accurate values of the integral. This is common practice in some cases but does not apply to oscillatory integrands because the value of the integral is highly sensitive to the integrand itself. A discussion of this has been added to the manuscript. Regarding the relative error compared to samples, we have added this to the existing figures.
>
> 5. Inductive bias of the model does not seem to be correct.
>
>    __Response:__  We agree that option 2., noted by the referee, appears phenomenologically consistent, as opposed to the approach of this manuscript. However, preliminary results showed that this does not improve our results despite it not having the most appropriate inductive bias. Further, all integrals end up being nonlinear approximates in both cases. We have not fully reformulated our manuscript, though agree with the referee in spirit that Option 2 makes more sense from a high-level point of view.
>
> 6. Inaccuracies.:
>     Below is a list of some minor problematic parts.
>     [...]
>
>     __Response:__ We have corrected and addressed all the points in the revised manuscript! We thank the referee for the sharp eye and thorough reading of our paper.
>
> 7. Not reproducible
>
>     __Response:__ We planned on providing our code but were unsure how to do it without breaking the double anonymity that TMLR requires. Our code is already in a private GitHub repository under the MIT license and is fully documented. We will, of course, make it public upon publication (should the manuscript be published).

---

> ### Comment · Reviewer_Eijb · 2024-03-11
>
> I want to thank the authors for providing a revised version and comments that address most of my points. In general, I am in favor of accepting the manuscript to TMLR.
>
> However, there are still a few questions that I want to clarify. These questions are listed below:
> 1. It is still not entirely clear to me what kind of functions are used.
>
>     a. For example, in the first line of Table 1, one can find Bessel$(k,v)$ defined as $f(\\cos(kx))J\_v (\\nu, x)$. I can not find what is $f$.
>
>     b. Similarly, in the legend of Figure 3a this $f(\cos(x))$ is repeated.
>
>     c. In Figure 3b, the Evan-Webster-1 differs from the function with the same name given in the table.
>
>     d. In Figure 5 one can find graphs of functions $\\sin(5x)$ and $\\sin(10x)$ that are for some reason presented as $f(\\sin(5x))$, $f(\\sin(10x))$. What is $f$? Perhaps, authors meant $f(x) = \\sin(5x)$, $f(x) = \\sin(10x)$
>
> 2. Still, the baselines are weak. My suggestion was to include specialized methods designed for highly oscillatory integrals. Authors commented
> > We also note that while some approaches have been proposed for a class of integrands, they are not fully general nor provide code for comparison. They also do not measure FLOP efficiency, which is of critical interest when computing an integral millions of times (or perhaps many more!) in such a many-query setting.
>
> which I find frustrating. First, in these articles, authors do measure FLOP efficiency: this is done by comparing the number of function evaluations. Second, although I agree that it is harder to find off-the-shelf integrators for highly oscillatory functions, I disagree, that this is impossible. (Moreover, I believe that it is the responsibility of the authors to *implement* them if they are not available.) For example, here is a discussion on the implementation of the Levin method in Julia https://discourse.julialang.org/t/rfc-ann-oscillatoryintegralsode-jl-levin-method-ordinarydiffeq/55601.
>
> I understand that this can be not straightforward, and it is not clear whether it will provide more insight into the method presented in the current contribution, so I leave this question at the discretion of the authors and the action editor.
>
> 3. I still believe that it would be interesting to look at the generalization study. Note, that it is not entirely true that neural networks are bad extrapolators. One may argue that all the neural networks do is an extrapolation https://arxiv.org/abs/2110.09485. Since the standard guidelines are absent, it is not clear how this study should be conducted. So, I do not insist on including it in the current version of the paper.
>
> 4. I appreciate authors include applications based on the Rayleigh–Plesset equation. I suggest providing more discussion on what this equation means (the part about applications/origin in physics) and what is the end of it. Appendix A3 is hard to read in its current form. An unprepared reader is caught off guard by alienated terminology and the mere precedes of second-order ODE. To help the reader, I suggest to add more details that can answer the following questions:
>
>     a. How is the ODE related to the integration problem considered in the article? Is there a way to recast this problem as an integral, and if so, how to derive it?
>
>     b. What is the meaning of variables in the ODE? What is the physical situation? What is the initial bubble radius? What is the ambient pressure?
>
> 5. To share the code I suggest trying the following popular tool often used to anonymize GitHub repositories https://anonymous.4open.science
>
> To summarise, I believe that subject to minor corrections suggested in this comment, the manuscript is suitable for publication in TMLR.

---

> ### Author Response · Authors · 2024-03-16
> **Response to response of reviewer Eijb**
>
> We thank the referee for his positive comments and appreciate the encouragement for acceptance. We respond below and have made many manuscript revisions.
>
> 1. It is still not entirely clear to me what kind of functions are used.
>  a.  For example, in the first line of Table 1, one can find Bessel(k,v) defined as [...] I can not find what is f.
>  b. [...]
>
> __Response:__
> These sub comments are in the same vein and point to insufficient care given to a few aspects of our notation. Of course, we have gone through the manuscript and updated it everywhere to be clear and consistent, going beyond the inconsistencies found by the referee. We thank the referee for the sharp eye.
>
> 2. Still, the baselines are weak. My suggestion was to include specialized methods  [...]
>
> __Response:__
> We think we better understand the referee's original comments now.
>
> Regarding using oscillatory integral-specific baselines: We agree that function evaluations can serve as a measure of computational cost, though there is often more involved in using Levin-type methods for general integrands than the formulation given in the original Levin (2006, 2007) papers. For example, one has to find a suitable basis for the integrand. In some cases this is straightforward: Levin himself does this on his test cases. However, if one does not know the appropriate oscillatory basis functions for integration, then the method quickly becomes complex, and function evaluations are likely not a suitable surrogate for FLOPs per integral evaluation. I am less familiar with the Julia package, though it appears suitable only when one manually specifies the basis.
>
> It is unclear to us how one could apply this strategy in practice (though we would be curious to learn if others have done this, it seems very challenging!). __I do know that__ Mathematica implements the Levin method, automatically attempting to find a suitable oscillatory basis for integration. It _seems_ to do a good job, and I believe one _might_ be able to extract the number of function evaluations from Mathematica. However, this is a closed source, and the methodology it uses to pick the oscillatory basis is hidden (as is the basis it actually picks!). It is also unclear how much evaluation must be done before one settles on that basis. Overall, it is broadly unclear to the authors how we could construct a truly representative baseline here without introducing significant confusion in the manuscript.
>
> While we have pushed back a bit on the referee's request, __we agree__ on what appears to be the broader point: That an oscillatory integrand-specific comparison could be an important point of contact for the reader. We believe this is not feasible in the scope of this manuscript (see above!), but entirely agree on the _broader idea_. There are numerical methods for highly oscillatory integrands and dismissing them so quickly is doing a disservice to those methods and perhaps unfairly props up our method. It is __possible__ that an integrator for specific highly-oscillatory integrands _would be_ more FLOP-efficient than using a neural network, it just seems unlikely to us that this could be the case in the many-query setting.
>
> In light of this, we have revised the manuscript again to __strongly hedge our results and their likely best practice use-cases__, noting all of the above points that our method turns on. Thank you for the pushback, referee Eijb.
>
> 3. I still believe that it would be interesting to look at the generalization study. [...] So, I do not insist on including it in the current version of the paper.
>
> __Response:__
> We agree with the referee here. We added a discussion that states our integrand-type out-of-training generalization is, at least in the cases we checked, not FLOP-efficient. The quantitative results do not appear to convey any information other than this statement and some elaboration upon it, so we did not include it in the second revision.
>
> 4. I appreciate authors include applications based on the Rayleigh–Plesset equation. I suggest providing more discussion on what this equation means (the part about applications/origin in physics) and what is the end of it. Appendix A3 is hard to read in its current form. An unprepared reader is caught off guard by alienated terminology and the mere precedes of second-order ODE. To help the reader, I suggest to add more details that can answer the following questions: [...]
>
> __Response:__
> Agreed. We have significantly expanded this section to make it as clear as possible to readers unfamiliar with our terminology, including the suggestions by the referee.
>
> 5. To share the code I suggest trying the following popular tool often used to anonymize GitHub repositories [...]
>
> __Response:__
> Amazing tool! I should have known about this. Please find our code here: https://anonymous.4open.science/r/oscilNN-D7FF
>
> Note that it will not work out of the box since so much has been automatically anonymized.

---

### Review · Reviewer_Ua35 · 2024-02-19

**Summary Of Contributions:**

The paper studies the efficiency of neural-network-based integrators compared to classical numerical procedures to integrate oscialtory one-dimensional functions. The idea consists simply in training multi-layer perceptrons to return accurate integrals of functions given as inputs a set of function values at given quadrature points. The authors delineate the potential of neural-network-based methods for highly oscillatory functions in terms fo flops and also point out potential limitations of the approach.

**Audience:**

Yes

**Broader Impact Concerns:**

Unforeseen.

**Claims And Evidence:**

Yes

**Requested Changes:**

**Desired changes**:
- Add comparisons to earlier work.
- While FLOPs are taken into account, a discussion about memory costs would be relevant. In particular it is not clear how to use the proposed approach if the network is trained on, say, [0, 1], and the function needs to be integrated on [10^3, 10^6].

**Suggested changes**:
- Present alternative architectures:
  - Change the ReLU to e.g. the GeLU or other activation functions?
  - Use convolutional layers or event transformers?
  Overall as the idea is simple, its execution could be more extensive.

**Minor presentation details**:
- In the experiements section, the forms of the function considered should appear as close as possible to the paragraph 3.1.1 training data.
- A full pipeline explained in e.g. pseudocode or with a figure needs to be added to understand the methodology more easily than by reading the text.
- Use a formula in the table, or anything that speaks more than "alpha". Also why is alpha normalized by the performance of the network and not the performance of the classical method?

**Strengths And Weaknesses:**

**Strengths**:
- The idea is very simple but worth a careful examination. The authors take care of analyzing the benefits of their idea in terms of practical considerations, namely FLOPs for a given accuracy.
- The paper delineates clearly some potential limitations of the approach which can help the practitioner decide which method to use.
- The methodology is relevant and rather well-explained.

**Weaknesses**:
- There is a lack of comparison to other methods. Since the idea is particularly simple, it is important to study alternatives that have been proposed in the literature. The argument against previous methods are clear (lack of expressivity of cosine activation for example in Y. Xu and J. Li's paper). However it'll be better to have experiments justifying such claims.
- Again while the methodology is simple, there is a also a technological opportunity to seize here that the authors could develop further. For example, it would be relevant to have access to the code of the authors to either use it as an alternative to classical integrators or to build one's own neural network to get trained integrators.
- Overall the paper may be better formatted, see some suggestions below (the current manuscript is ok though and the suggestions below are just suggestions up to the authors.).

---

> ### Author Response · Authors · 2024-03-02
> **Response to reviewer Ua35**
>
> ## Weaknesses
>
> 1. Lack of comparison to other methods.
>
>     __Response:__ This is discussed below in the "Suggested Changes" part of our response. Thanks!
>
> 2. "[...] it would be relevant to have access to the code of the authors to either use it as an alternative to classical integrators or to build one's own neural network to get trained integrators."
>
>     __Response:__ Thank you, we agree! We planned on providing our code, but we were not sure how to do it without breaking the double anonymity that TMLR requires. Our code is already in a private GitHub repository under the MIT license and is fully documented; we hope others build upon it and test it on their applications!
>
> 3. "Overall the paper may be better formatted [...]"
>
>     __Response:__ We agree. We have made changes in the revised manuscript following the referee's ideas.
>
> ## Desired changes
>
> 1. Add comparisons to earlier work [this corresponds to Weakness 1).
>
>     __Response:__ We could not find comparisons to earlier work that explicitly discussed FLOP efficiency in the 1D context, which is especially relevant to many physical problems, in comparison to, e.g., 8D integrands. Even the works that volley neural network ideas do not provide code (at least that we could find) to replicate and compare against their results.
>
> 2. While FLOPs are taken into account, a discussion about memory costs would be relevant. In particular it is not clear how to use the proposed approach if the network is trained on, say, [0, 1], and the function needs to be integrated on [10^3, 10^6].
>
>     __Response:__ We have added Appendix A1 to address this comment. In short, the memory burden carried by such a neural network is very small (O(100) bytes). This could be reduced via lower floating point precision, though unnecessary. The constraint of FLOP efficiency necessitates small neural networks, so the memory footprint is not very large. Regarding the comment on how one could deal with computing an integrand on a different domain that was present in the training set, one would transform the quadrature points to the desired domain via shifting (addition/subtraction), then stretching/shrinking (multiplication), then compute the integrand. This is standard practice for, for example, Gauss quadrature rules, for which the quadrature points are already tabulated in a library on a pre-determined domain like [0,1] or [-1,1]. The same strategy applies here but does not increase costs.
>
> ## Suggested changes
>
> 1. Present alternative architectures:
>     * Change the ReLU to e.g. the GeLU or other activation functions?
>
>         __Response:__ ReLU is selected to reduce computational expense. The network is shallow, and the final integrals are always positive. Thus, it does not appear that GeLU offers a meaningful advantage.
>
>     * Use convolutional layers or event transformers? Overall as the idea is simple, its execution could be more extensive.
>
>         __Response:__ Similar to the above point, we had to focus on simple architectures to keep the FLOP count low. CNNs and transformers offer accuracy advantages, but the inference cost is disproportionately higher. Thus, we did not explore this strategy. It would make much sense in many _other_ contexts, though.
>
> ## Minor presentation details
>
> 1. In the experiments section, the forms of the function considered should appear as close as possible to the paragraph 3.1.1 training data.
>
>     __Response:__ We agree. We have adjusted our wording. Now, all the functions are given with inputs in the same range of (2^0 to 2^13), i.e., the number of separate inputs $x$ on the domain [a,b].
>
> 2. A full pipeline explained in e.g. pseudocode or with a figure needs to be added to understand the methodology more easily than by reading the text.
>
>     __Response:__ The pipeline is as simple as it seems: a feed-forward neural network is trained to approximate an integral via input data at fixed points. Figure 1 shows this, though we added further description in the caption to make it clearer. We hope the reviewer agrees this to be appropriate.
>
> 3. Use a formula in the table, or anything that speaks more than "alpha". Also why is alpha normalized by the performance of the network and not the performance of the classical method?
>
>     __Response:__ The performance indicator $\alpha$ is defined in an equation closely snugged against the table that references it. This is also described in the table caption. We have adjusted the table caption to make this more clear. The caption also explicitly references the equation. The $\alpha$ metric is normalized by the performance of the network as it represents the gain in number of FLOPs after using the network. We also wanted to demonstrate the performance of the network did indeed perform better. If we had chosen to normalize the performance of the classical method, then it would have been the opposite.

---

### Author Response · Authors · 2024-03-02
**Broad comments, responses, and appreciation!**

We thank all of the referees for their time and keen eye in the review of our manuscript. We appreciate that their appears to be consensus that the method can be quite scientifically useful. The main concerns appeared to arise over (i) the baseline for our comparisons (like the midpoint method, which could appear as a straw-man comparison), (ii) the size of the training set and its use, and (iii) reproducibility. These are addressed specifically and at length below, but we would like to address these common concerns briefly below.


(i) We have added higher-order accurate baseline approximations from classical integrators. However, these perform nearly the same in all cases when compared to the other classical integrators. This is because, for highly oscillatory integrands, none of the integrators reach their asymptotic convergence limits, and it would be prohibitively expensive for them to do so. This is actually one of the main motivations of this work.

(ii) We have expanded the training set both in scope by including more integrand families that are qualitatively different in scope. Additional figures and figure revisions have been added in light of this. We have also clarified a mistake in our notation in Table 1: We actually train on a range of parameters that classify the integrand, rather than a fixed integer set of them. This was a mistake on our part and is now fixed, but the previous results hold.

(iii) We wanted to include the code, but were not sure how to best do this without breaking anonymity. The code currently lives in a private GitHub repo. and is fully documented. It is MIT licensed, so anyone can use it and create their own versions if they like. The repository will be switched to public upon publication (should the manuscript be published, of course!).

---

### Decision · Action_Editor_Rwcr · 2024-03-28

**Recommendation:** Accept as is

**Comment:**

I agree with the reviewers that the claims are sufficiently interesting and supported well enough to justify accepting the paper for TMLR.

**Audience:**

It will be of interest primarily to a small subset of the TMLR audience.

**Claims And Evidence:**

The paper evaluates whether neural networks can be used to integrate functions with high oscillations.

The reviewers agree that this in an interesting task to pursue and while the setup is simple the majority agrees that the provided evidence for the proposal is sufficient, especially given the changes introduced during the rebuttal.

Therefore I recommend acceptance.